# The Impact of Social Comparisons More Related to Ability vs. More Related to Opinion on Well-Being: An Instagram Study

**DOI:** 10.3390/bs13100850

**Published:** 2023-10-17

**Authors:** Phillip Ozimek, Gabriel Brandenberg, Elke Rohmann, Hans-Werner Bierhoff

**Affiliations:** Department of Social Psychology, Ruhr-Universität Bochum, 44801 Bochum, Germany; gabriel.brandenberg@rub.de (G.B.); elke.rohmann@rub.de (E.R.); hans.bierhoff@rub.de (H.-W.B.)

**Keywords:** Instagram, social networking sites (SNSs), social comparison orientation, well-being

## Abstract

Social networks are gaining widespread popularity, with Instagram currently being the most intensively used network. On these platforms, users are continuously exposed to self-relevant information that fosters social comparisons. A distinction is made between ability-based and opinion-based comparison dimensions. To experimentally investigate the influence of these comparison dimensions on users’ subjective well-being, an online exposure experiment (*N* = 409) was conducted. In a preliminary study (*N* = 107), valid exposure stimulus material was selected in advance. The results of the main study indicated that the exposure to ability-related social comparisons in the context of social media elicited lower well-being than exposure to opinion-related social comparisons. The theoretical and practical implications of this study consist of including the findings in clinical settings, e.g., affective disorder therapy, and the identification and reduction of ability-related content on social networking sites (SNSs). Future work should include assimilation and contrast effects which might interact with social comparison orientation and well-being.

## 1. Introduction

Facebook, X, and Instagram collectively boast up to 2.7 billion active users per month; so-called social networking sites (SNSs) have been growing in popularity for two decades [1]. On SNSs, users create their own electronic profile and can interact with other users in a variety of ways. This results in new forms of interaction that have also aroused research interest [2]. Numerous studies have already elaborated on the motivation to use SNSs and observed both positive and negative effects of the use of SNSs [3,4,5,6,7]. SNSs are about people. Therefore, the use of SNSs is intrinsically linked to the elicitation of social comparisons. They are frequently triggered when users are confronted with information about another individual that is related to their self [8]. Such information is omnipresent on SNSs, which is why social networks provide such a powerful platform for the elicitation of social comparisons online. A question of particular relevance, both personally and societally, is what impact do these comparisons have on users’ well-being? This question was investigated in our study.

### 1.1. Social Networking Sites (SNSs)

Ellison and Boyd [9] defined SNSs as networked communication platforms where users create uniquely identifiable profiles that may include their own content as well as content from other users. Diverse content can be both consumed and produced. Interactions with this user-generated content are also possible. In addition, connections to people published by users can be accessed and viewed by other users. In general, SNSs differ in their designs and functions. The most popular functions include sending private messages, liking posts, uploading one’s own photos, and interacting with posts from other users [1]. Options such as creating group chats, joining group pages, or creating events are also available on many SNSs.

Depending on the available functions, SNSs can differ accordingly in persistence, connections, visibility, and editability [10]. In addition, each SNS has a different focus. For example, Instagram is an image-based social network, while X is predominantly text-based.

The platform Facebook currently has the highest number of users, with over 2.7 billion monthly active users [1]. However, the most intensively used SNS is Instagram [3]. Instagram is characterized as an image-based SNS due to the fact that photos can be edited and shared on one’s own profile [11]. In addition, moments from everyday life can be shared, which are visible for 24 h on the so-called “story”. It is also possible to privately send photos, videos, and messages to other users. The number of monthly active Instagram users doubled from 2013 to 2018, so that Instagram now has around two billion active accounts [1]. This development also aroused research interest regarding Instagram, which caused a rise in scientific papers on the subject. Since SNSs offer a new type of communication and interaction due to the functions mentioned [2], more intensive research must be conducted in the future to determine what the offline behavior of users looks like, what the motivation for use is, and how this can be compared with online behavior. The described designs of SNSs require, among other things, that users are confronted with the posted content of other users, so that through this social information social comparisons automatically take place.

### 1.2. Social Comparisons

Festinger [12] postulated in his theory of social comparison that comparing the self with others is a basic human need. The standard of comparison are people that are perceived as similar in relevant dimensions such as performance, success, and health. The theory focuses primarily on the motivational reasons for comparison and indicates that despite the existence of objective standards, subjective information gained through social comparison still has an influence. These comparisons take place whenever information about other individuals is available [8]. Information about the self can be gained from those comparisons and self-evaluation occurs. From a systematic point of view, three different comparison directions are distinguishable, i.e., lateral, upward, and downward. Therefore, social comparisons refer either to similar, superior, or inferior others. These comparison processes can take place either consciously or unconsciously (cf., [13]).

Individuals compare themselves to others to assess their own abilities or opinions in relation to the comparison person’s [14]. SNSs provide platforms on which social comparisons are enabled, as self-referential information is continuously presented and retrievable (cf., [8,15]). Instagram, as an image-based medium, provides quick and easy ways to access millions of profiles and use them to collect social comparison information, and allows people to present themselves [16,17,18]. Accordingly, Instagram can meet the basic needs for the social comparison described above.

Sheldon and Bryant [7] generally identified four motives for using Instagram; in addition to photo or video documentation, creativity, and coolness, observing others is an important motive for use. According to Mussweiler et al. [8], social comparison automatically results from observing others. It has also already been shown for Facebook that the need to compare oneself functions as a relevant motive for use. Thus, in addition to the need for belonging and the need for self-presentation, the need to compare was added [14,15,19,20].

The Social Online Self-Regulation Theory (SOS-T [15]) embeds the motives for using SNSs in an overarching framework and considers SNSs as means for the purpose of self-regulation. Self-regulation describes a process by which one’s thoughts and actions are controlled to achieve positive and avoid negative end states [21]. This process is mostly unconscious and occurs mostly automatically [22]. It is assumed that people have motives in terms of higher-level goals, which in turn activate specific goals that can be achieved by different means [23,24].

The numerous opportunities for social comparison presented on SNSs are likely to influence users’ subjective well-being. For example, the inference of one’s own superiority is likely to enhance subjective well-being, whereas the inference of one’s own inferiority is likely to reduce subjective well-being.

### 1.3. Subjective Well-Being, Social Comparisons, and SNSs

In general, different results of social comparisons may arise. The construct of subjective well-being (SWB) turns out to be particularly important in this context. SWB constitutes a multidimensional construct which is divided into an affective and a cognitive component [25].

The affective component includes the positive and negative affect, whereas the cognitive component refers to life satisfaction and evaluation of the self. In this context, affect refers to a persistent state that manifests itself in short-term moods and emotions and is influenced by internal and external factors [26]. Life satisfaction refers to a comprehensive cognitive evaluation process of quality of life based on comparing one’s own situation with an adequate situational standard [27]. Our approach is based on the tripartite model of happiness [28] which combines measures of life satisfaction, and positive and negative affect.

In addition, we included self-esteem as an indicator of subjective well-being which constitutes a positive characteristic of happy people and connects happiness with flourishing and positive mental health [29]. High self-esteem is defined as the positive evaluation of one’s own person and reflects positive thoughts toward the self (in contrast to negative thoughts toward the self [30]). In general, stable global self-esteem, as the overall evaluation of the self, is contrasted with the state of self-esteem, as momentary self-esteem varying depending on time and situation [31]. This research focusses on global self-esteem.

In general, social comparisons elicited on SNSs tended to reduce subjective well-being [6,14,32,33,34,35,36,37,38]. One possible explanation is that self-presentation on SNSs is excessively positive and idealized [39]. In effect, users frequently compare their real selves with the ideal selves of others, which may result in a feeling of inferiority. As a consequence, the elicitation of upward comparisons are likely to facilitate negative feelings [15,40,41]. Specifically, initial evidence has been found that social comparisons generated on Instagram have negative repercussions on well-being [17,18,42]. In agreement with these results, further studies, which were conducted recently, revealed a negative association between social comparisons on SNSs and subjective well-being [43,44].

This pattern of results was modified after taking the distinction between active and passive use of social media into account. Whereas active users communicate directly with others (e.g., posting comments, chatting, uploading content), passive users merely consume the content elaborated by others without interacting with them (e.g., reading comments, viewing profiles [45]). Taking this distinction into account, it became apparent that active use was associated with higher subjective well-being, while passive usage was associated with lower subjective well-being [46,47,48].

Another important distinction refers to ability- and opinion-oriented comparisons [41]. This distinction was originally introduced by Festinger [12] in his groundbreaking publication on social comparisons, who contrasted opinions and abilities by pointing out that opinions possess no objective basis of evaluation whereas abilities are measurable in terms of objective performance criteria. Festinger [12] postulated that social comparisons with others who are expected to be close to one’s own position result in stable evaluations of opinions and abilities. In addition, he postulated that abilities elicit a pursuit to become better, distinguishing them from opinions: a unidirectional upward orientation.

In the same vein, Park and Baek [41] focused on the distinction between opinions and abilities in the context of upward and downward social comparisons. Following the Social Comparison-Based Emotions model [49], they distinguished between upward assimilative emotions (optimism, inspiration), upward contrastive emotions (envy, depression), downward assimilative emotions (worry, sympathy), and downward contrastive emotions (schadenfreude, pride). Specifically, their results indicated that a high ability-oriented social comparison orientation led to less subjective well-being based on upward comparisons (but not based on downward comparisons). In addition, a high opinion-oriented social comparison orientation led to more subjective well-being because of upward comparisons. Their research, which integrated several emotions, led to complex results. But a main path of influence was revealed from ability-based comparisons via envy/depression (positive association) to satisfaction with life (negative association). Therefore, ability comparisons considerably increased envy and depression, which in turn strongly reduced satisfaction with life. This path was much stronger than the other paths considered (e.g., opinion-based comparison via worry/sympathy (positive association) to satisfaction with life (positive association)).

These results indicate that the distinction between ability-oriented and opinion-oriented social comparisons is promising in terms of satisfaction with life. The implications of this distinction were investigated in this research by employing an experimental design instead of a correlative path model. Therefore, we focused on causal analysis within the SNS context.

### 1.4. Hypotheses

Two hypotheses were pursued which are interrelated, referring to the structure of the dependent variables on the one hand and to the ability/opinion distinction on the other hand.

In correspondence with the tripartite model of happiness [28] and conditions of mental health as outlined by Seligman [29], high subjective well-being should be given if positive affect is high, negative affect is low, satisfaction with life is high and self-esteem is also high. In accordance with this taxonomy, we proposed that the indicators of subjective well-being are significantly correlated. Thus, satisfaction with life, self-esteem, and positive affect should be correlated positively whereas the three indicators should be negatively correlated with negative affect (**H1**).

To investigate the relationship between ability/opinion based social comparison and subjective well-being in terms of affect, satisfaction with life, and self-esteem we employed an experimental design. Note that Park and Baek [41] found that ability-based social comparison orientation was strongly related to decreased subjective well-being. Is this pattern of results replicable experimentally and does it generalize across different measures of subjective well-being which cover both self-esteem and satisfaction with life?

The experimental manipulation was realized through the deployment of two versions of social comparison information in the context of SNSs. In response to Park and Baek’s results [41] the following hypotheses were outlined: an exposure to ability-related social comparisons leads to lower subjective well-being in terms of lower positive affect (**H2a**)**,** higher negative affect (**H2b**), lower life satisfaction (**H2c**), and lower self-esteem (**H2d**) than an exposure to opinion-related social comparisons.

## 2. Method

### 2.1. Research Design

The present study adopted an experimental exposure paradigm in an online format (cf., [50]). Participants were randomly assigned either to ability-based or to opinion-based exposition groups. An online setting was chosen for the implementation of the experiment, because it is highly economical (cf., [51]) and consistent with digital consumption of SNSs. Data for the preliminary as well as the main study were collected between April 2019 and October 2022 at the Ruhr University Bochum well as via social media.

### 2.2. Preliminary Study

To generate suitable exposure material in our main study, a preliminary study was conducted. Data were collected through an online questionnaire distributed via Qualtrics. Anyone who was of legal age (above 18 years old) and had an Instagram account was asked to participate. A total of 107 persons participated in the preliminary study, with an overrepresentation of female respondents (82.2%). The mean age of the participants was 23.92 years (*SD* = 4.50) and their average Instagram usage time was between 30 and 90 min per day.

The pre-study included ten stimuli that were intended to expose respondents to ability- or opinion-based social comparisons. The presentation of the stimuli was random to avoid sequence effects. In general, the stimuli resembled an Instagram profile (cf., Section A.1, Section A.2, Section A.3 and Section A.4). The stimuli were photos, either self-taken or license-free, tagged with appropriate hashtags to increase authenticity. A gender-neutral user profile named “lighthouse_xx” was used for this purpose. After each of the stimuli were presented, participants were asked whether the post was more likely to reflect opinions or abilities to ensure a high face validity. Note that Festinger (1954, p. 118) wrote that “the clarity of the manifestation or performance can vary from instances where there is no clear ordering criterion of the ability to instances where the performance which reflects the ability can be clearly ordered”. Thus, there is a wide range of so-called “performance criteria”. For this, we choose stimuli where “ability” through activity is the focus, showing a high face validity. Finally, participants could optionally express criticism in a free text field.

The basis for the stimulus selection for the main study was which stimuli most closely reflected ability and opinion, respectively. Therefore, mean ratings were considered. Values close to 0 indicated ability-related content and values close to 1 indicated opinion-related content (see Appendix B). Five stimuli each, which on average were closest to the intended content (ability or opinion, respectively), were selected for use in the main study (see Section A.1, Section A.2, Section A.3 and Section A.4).

### 2.3. Main Study

#### 2.3.1. Sample

To determine the required sample size, a sample size design was first conducted using G*Power (version 3.1.9.6; [52]) assuming mean effect size and a significance level of 5%. Since the *t*-test is the most complex statistical procedure used in analysis, a minimum sample size of 302 was required. 415 respondents finally participated in the study. After six respondents were excluded due to incomplete data sets, 409 participants were included in the final sample. Of the participants, 338 were female (82.6%), 63 were male (15.4%), and 8 (2.0%) were gender diverse. The average age was 24.31 years (*SD* = 5.31). Most participants reported having a high school degree (210; 51.3%) followed by an academic degree (155; 37.9%), a secondary school certificate (42; 10.3%), or no school diploma (2; 0.5%). The daily time spent on Instagram averaged between 30 and 90 min. Participants on average reported themselves as following 319.33 (*SD* = 201.05) Instagram profiles and to be, on average, followed by 299.23 (*SD* = 247.02) Instagram profiles.

#### 2.3.2. Procedure

The main study was also conducted on Qualtrics. The recruitment of participants was achieved via a snowball sampling technique. The inclusion and exclusion criteria were the same as in the preliminary study.

The study started with participant information and informed consent, which was followed by exposure instructions. Participants were shown, in a randomized manner, four of the selected five stimuli in the ability- or opinion-related photographs category. They were randomly assigned to the experimental groups (*N*_ability_ = 198, *N*_opinion_ = 211). Using Chi-Square tests we checked for significant differences in the groups with respect to demographic (i.e., gender, age, educational degree, number of followers and followees as well as mean time spent on Instagram). However, no significant difference occurred, all *p* > 0.05. To ensure intensive engagement with the stimulus material, a free-text field with appropriate instructions for engagement with the material was inserted after each stimulus. Participants had to answer (in at least 30 characters) how they would describe the person who presumably posted the picture. In addition, a manipulation check was included by asking the participants after each stimulus whether what they saw was more related to abilities or opinions. After the manipulation of the independent variable was completed, the dependent variables were collected.

#### 2.3.3. Measures

The dependent measures included three questionnaires: (positive and negative) affect, life satisfaction, and self-esteem. Life satisfaction represents the stable component of cognitive well-being. The three scales were presented in a randomized order to reduce the likelihood of systematic bias due to sequence effects. This was followed by inquiry into demographic data (gender, age, highest education, daily Instagram usage time, and number of followers).

***Affect.*** The affective component of subjective well-being was assessed using the German short version of the Positive and Negative Affect Schedule (PANAS; [53]) by Randler and Weber [54]. The scale assesses positive (e.g., [I feel] “active”) and negative (e.g., [I feel] “angry”) affect on a five-point Likert scale (1 = very little/not at all to 5 = extremely) with five items each. The scale showed an acceptable internal consistency of α = 0.74–0.79 for positive affect and α = 0.66–0.70 for negative affect [54]. In the present study, acceptable reliability was also found (α_positive_ = 0.69, α_negative_ = 0.79).

***Life Satisfaction***. A standard measure of life satisfaction was employed. The German version of the Satisfaction With Life Scale (SWLS) by Glaesmer et al. [55] is based on the original scale by Diener et al. [27]. Five items (e.g., “In most areas, my life matches my ideal”) capture personal quality of life on a seven-point Likert scale (1 = I strongly agree to 7 = I strongly disagree). The scale achieved excellent internal consistency with an α of 0.92 [53]. High internal consistency (α = 0.87) was also achieved in the main sample.

***State Self-Esteem***. State self-esteem reflects cognitive well-being. It was assessed using a revised German version of the State Self-Esteem Scale (SSES), originally developed by Heatherton and Polivy [56] (SSES-R; [31]). Fifteen items are answered on a five-point Likert scale (from 1 = strongly disagree to 5 = strongly agree). Three subscales, each with five items, are distinguished: performance self-esteem (e.g., “I have confidence in my abilities”), social self-esteem (e.g., “I care about the impression I make”), and appearance self-esteem (e.g., “I think I look good”). In addition, an overall scale including the 15 items is possible. We report high internal consistencies for all subscales (α_performance_ = 0.80, α_social_ = 0.87, α_appearance_ = 0.88, and α_total_ = 0.90). The current sample displayed high internal consistency in the SSES-R (α = 0.90 for the combined scale across performance, social, and appearance).

## 3. Results

### 3.1. Descriptive Statistics and Intercorrelations

Firstly, descriptive data (Table 1) and intercorrelations (Table 2) of the measures are reported. In general, participants reported medium ratings. An exception is the PANAS_negative_ scale, because the ratings for negative affect were very low in general.

H1 focused on the correlation pattern of PANAS, SWLS, and SSES. In accordance with previous research, the two subscales of the PANAS correlated significantly negatively. High positive affect implied less negative affect and vice versa. In addition, agreement with the hypothesis of significant positive correlations between PANAS_positive_, the SWLS, and the SSES emerged, whereas PANAS_negative_ displayed, as expected, negative correlations with the SWLS and SSES. The highest (positive) correlation was recorded between SWLS and SSES, indicating 43% of the common variance. Even the lowest correlation between PANAS_positive_ and PANAS_negative_ accounts for 12.9% of the common variance detected via R^2^. As expected, high subjective well-being is consistently captured via PANAS, SWLS, and SSES.

### 3.2. Manipulation Check

The manipulation check examined the extent to which the manipulation achieved the intended effect, i.e., whether the respondents made the correct assignment when asked after each stimulus presentation whether its content was related more to abilities or more to opinions. Overall, the stimuli were correctly assigned. Only three stimuli were categorized incorrectly by some participants. The details of the manipulation check are summarized in Appendix C.

### 3.3. t-Test

The second hypothesis stated that an exposure to ability-related social comparisons leads to lower subjective well-being than an exposure to opinion-related social comparisons. H2 was examined based on the experimental design using *t*-tests for independent samples. Of the 409 participants, 198 were randomly assigned to EG_Ability_ and 211 to EG_Opinion_. There was only a marginally significant difference between EG_Ability_ and EG_Opinion_ with respect to positive affect (*t*(407) = −1.84, *p* = 0.067, *d* = 0.71), so hypothesis 2a had to be rejected. Negative affect, on the other hand, was significantly higher in EG_Ability_ than in EG_Opinion_ (*t*(382) = 2.09, *p* < 0.05, *d* = 0.75). This was consistent with hypothesis 2b. Also, there was a statistically significant difference in life satisfaction between EG_Ability_ and EG_Opinion_ in the expected direction (*t*(407) = −2.22, *p* < 0.05, *d* = 1.26). This result was consistent with hypothesis 2c. Finally, a lower self-esteem in the EG_Ability_ than in the EG_Opinion_ condition was observed. This difference was statistically significant (*t*(407) = −2.03, *p* < 0.05, *d* = 0.72) and was, therefore, consistent with hypothesis 2d. The results of these *t*-tests are graphically illustrated in Figure 1.

## 4. Discussion

A special feature of this research is that the between-subjects design of this study allows for a causal interpretation of results: ability-based comparison information caused a reduction in psychological well-being (compared with opinion-based comparison information). This experimental effect is replicated across several indicators of quality of life and therefore represents a viable result. The questionnaires employed to measure subjective well-being (PANAS, SWLS, SSES) are standard measures which demonstrated the considerable reliability and validity of previous research. This study enters new territory by demonstrating a reliable difference between social comparisons based on abilities and opinions (cf., [12]), respectively, which is relevant in applied settings of the online community in general and on SNSs in particular. In summary, ability-based social comparison information is potentially more damaging in terms of self-evaluation than opinion-based information.

Regarding the research hypothesis, results indicated, in agreement with H1, that the dependent variables positive and negative affect, life satisfaction, and self-esteem, are correlated significantly with each other. They represent different facets of happiness with negative affect focusing on the negative pole of subjective well-being, whereas positive affect, life satisfaction, and self-esteem represent the positive pole. These results correspond with earlier studies which revealed that positive affect and life satisfaction, which represent the affective and cognitive components of subjective well-being, are correlated with each other, whereas negative affect is a negative concomitant of happiness. This approach agrees with the widely accepted tripartite model of happiness [28] which combines life satisfaction, positive affect, and negative affect as correlated dimensions. In addition, self-esteem was included, which constitutes a positive characteristic of happy people, and connects happiness with flourishing and mental health [29].

In correspondence with H2, the results indicated that an exposure to ability-related social comparisons in the context of social media led to higher negative affect (H2b), lower life satisfaction (H2c), and lower self-esteem (H2d) than exposure to opinion-related social comparisons. For positive affect, however, only a marginally significant difference between the two experimental conditions occurred, which is why hypothesis H2a was only weakly supported.

The results, with respect to the hypotheses, were consistent with previous research by Park and Baek [41], but instead of being correlational, they were based on an experimental design of random assignment. Therefore, the negative consequences of exposure to ability-related information in contrast to opinion-related information, which were detected in this experiment, are likely to represent causal influences instead of mere associations. The interpretation is that ability-related comparison information weakens subjective well-being as measured by several indicators of quality of life. Note that the general trend of the results seems to indicate that ability-related information reduced subjective well-being. Therefore, the overall pattern of results indicated that H2 was mostly confirmed.

One factor which could contribute to the stronger impact of ability-related social comparison information on well-being (cf., H2) might be that ability-related social comparisons, in contrast with opinion-related social comparisons, are automatically instigated [8,16,18]. Further research is needed to cast new light on this issue, such as the additional inclusion of comparison direction (up or down) or the kind of comparison process (assimilation or contrast).

The results of this study confirm the research findings of Ozimek and Bierhoff [35], who in three investigations observed decreased self-esteem and higher depressive tendencies as a consequence of socially comparative activities on SNSs. Furthermore, the authors revealed a systematic association of passive SNSs use with higher depressive tendencies, mediated by higher ability-related comparison orientation and lower self-esteem.

## 5. Theoretical and Practical Implications

The findings obtained are pertinent to theories such as social comparison theory [12], social comparison orientation [55], and also to the Social Online Self-Regulation Theory (SOS-T, [15]). Our findings expand Festinger’s [12] social comparison theory, in that social comparison occurs in situations where subjective information is salient. In line with his theory, self-evaluation also happens in online contexts. With regard to social comparison orientation, our preliminary but also main study support the well-established distinction in social comparison orientation [57] and the distinction between ability- and opinion-based comparisons. According to the SOS-T, SNSs represent a means to achieving individual goals. The extent to which they actually contribute to the achievement of the target state is not relevant for the choice process. Only the individual assessment of the means-goal link influences behavior [15]. Happiness and well-being could be seen as a global goal of people’s lives. However, our study has shown that practical implications follow from the particular orientation described in the practical implications.

Previously, alternative interpretations based on other theories have also been considered. For example, objective self-awareness theory [58] postulates that self-confrontation or self-reflection directs attention from the environment toward the self. An individual’s attention is then turned inward, becoming an object of one’s own awareness. Under conditions of increased self-awareness, the intensity of affects tends to be increased. Furthermore, increased self-awareness enhances the awareness of discrepancies between the real and imagined self. As a consequence, individuals tend to consider themselves more negatively when a negative discrepancy between the real and imagined self is perceived. In the SNSs context, this would reinforce the negative effect on well-being of comparing the real self with idealized selves on SNSs (cf., [39,40]). This assumption deserves further examination. However, objective self-awareness theory does not differentiate between ability-based and opinion-based comparisons, which was the focus of this research. Therefore, the theory must be refocused before it is possible to apply it to the contrast between ability-based and opinion-based comparisons.

With respect to application, the findings of this work are of practical relevance for clinical psychology. This study provides an initial starting point for developing recommendations for action for psychotherapists and clinicians in the treatment of people with mental illness. This study already gives a preliminary indication that the consumption of content on social media, which can trigger ability-based comparisons, can especially lead to negative consequences. An initial recommendation for action could be to use social media whose features favor opinion-based social comparisons, such as X, since only text-based communication is possible here. Nevertheless, further studies are necessary to further validate and specify these initial findings. As Park and Beak [41] have already shown in their correlative model, direction and psychological proximity (assimilation vs. contrast) are crucial in addition to the content of the comparison.

Additionally, whether SNSs use represents dysfunctional or functional means of self-regulation (cf., [15]) has corresponding implications for clinically relevant factors. Well-being plays an essential role in relation to psychopathological processes, especially with respect to affective disorders such as depression. Therefore, it is important to study the mechanisms and effects of SNSs use to facilitate more conscious goal-directed use that is not harmful to mental health, but beneficial for quality of life.

Thus, further studies should be devoted to the question of what can be done to make the consequences of using SNSs more positive. In the case of Instagram, there is already a so-called “Well-being Team” [11], which works on this issue. For example, it was established that photos digitally edited with Instagram would receive a license plate, which could reduce the negative impact of comparing real selves with idealized representations of people. In some cases, likes have also been turned off and alerts have been introduced when searching for hashtags on sensitive topics such as anorexia or depression, as well as pointing to sources of help. Within this framework, additional measures might be incorporated to reduce the negative consequences of using social network sites. For example, the relatively strong impact of ability-based comparison information on subjective well-being might be highlighted and provided with warnings for the users. This could be a starting point for developing guidelines for the ethical design of social media platforms.

## 6. Limitations

Although the preliminary study brought about an optimal selection of test materials in terms of abilities and opinions, the results of the present work should be understood in light of several limitations. First, the sample of this study was relatively homogenous with respect to age (i.e., young adults). Although SNS use decreases with increasing age [34], which is why most young adults are represented on SNSs, it would be desirable to include more older respondents in the sample. The sample is also unbalanced with regard to participants’ gender because significantly less men than women participated in the study. These biased sample characteristics limit the representativeness of the research reported here and attention should be directed to a balanced ratio of women and men in future studies. This also limits the study’s implications, since, for example, recommendations for clinicians based on our findings apply less to the general public and more to young women. However, the previous literature did not indicate any relevant effect on the results in this regard.

Second, the experimental design focused on EG_Ability_ and EG_Opinion_ groups, while an explicit control group was not included in the research design. Therefore, the experimental groups constituted base lines for each other. Although in general the use of an explicit control group is desirable, most of the time researchers compare two or more experimental groups with each other. Because the experimental groups adequately represented the research hypothesis, contrasting ability-based and opinion-based conditions, the design is suitable for the test of the experimental hypothesis. With respect to our implications, note that since we lack such a control baseline, it is unclear what the impact is when Instagram posts trigger no social comparison. This in turn limits the recommendations for action that can be developed, since we can only make relative statements about both types of comparison.

Third, this research exclusively focused on Instagram, which represents the fastest growing and most intensively used SNS [3,7]. Instagram is an image-based platform, which is important to consider when making generalized statements about SNSs. The results of the experimental study primarily apply to image-based SNSs. To that extent, our implications for clinicians are also limited to image-based SNSs. Since previous research tended to focus on the Facebook platform, the present findings extend the generalizability of previous findings by examining a different SNS.

Fourth, our research design was a reduced version of the research design employed by Park and Baek [41], because we only considered the comparison dimension and did not examine the Social Comparison Based Emotions model as a whole [49]. This limits the implications, in that there might be a gap between comparison orientation and well-being in which a differentiation of particular emotional responses right after stimulus exposition might explain the change in well-being and help with further practical implications. Social Comparison Based Emotions [49] differ in their direction: to oneself (e.g., optimism), focused externally (e.g., admiration), or dual in direction (e.g., inspiration). Although this theory has not yet been sufficiently validated, this distinction might help to further our understanding of this process. In our study we were interested in obtaining experimental results on the effects of ability-based and opinion-based social comparisons, respectively, on well-being. It seems hardly feasible to run a test of the complete Smith [49] model using a between-subjects experimental design. However, we point out a short framework for it down below. Note that Park and Baek [41] employed a within-subjects design. Nevertheless, replication based on the overall model, which is based on a between-subject design, seems to be within the scope of the exposure paradigm and should rather be explored in the context of longitudinal research. The comparison of our sample with the Park and Baek’s sample is instructive because Park and Baek employed a national sample of internet users which was representative of Korean internet users, whereas our study was based on a convenient sample of internet users. Therefore, our sample is not representative of German internet users. But it is hardly possible to implement an experimental design within a representative sample. Nevertheless, Park and Baek’s results [41] revealed, convincingly, that a focus on ability turns out to be detrimental (relative to the focus on opinion) in terms of consequences for psychological well-being.

The current study found significant differences in well-being depending on the elicitation of ability- and opinion-related comparisons, which were compatible with Park and Baek’s original results [41]. With respect to future research, the inclusion of further model components is desirable. To sharpen the assimilation vs. contrast effects on well-being, further research should validate Smith’s [49] model of Social Comparison Based Emotions in SNSs context. A similar stimulus design was used in this study to establish a 2 × 2 (assimilation vs. contrast; upward vs. downward comparison) within-between interaction design to experimentally verify the respective emotional reactions (before and after stimulus) to the four dimensions of the model. As a part of well-being [25], these emotional responses should be reflected in different measures of well-being. If the theory-implied emotions cannot be recovered, another exploratory approach could be to identify and allocate relevant short-term emotional responses as the basis for further research.

To holistically test various social comparison conditions on well-being, a large-scale study should consider (a) social comparison orientation (ability vs. opinion), (b) social comparison direction (upward vs. downward vs. lateral), and (c) the social comparison process (assimilation vs. contrast), i.e., in a 2 × 3 × 2 + 1 (control group) design. Furthermore, in a second step, the stability of these effects should be reflected by conducting a (or several, if investigated separately) long-term study with an ambulatory assessment design to minimize laboratory effects. In a third step, real behavioral data from an SNS system could help with fully evaluating the interdependencies.

## Figures and Tables

**Figure 1 behavsci-13-00850-f001:**
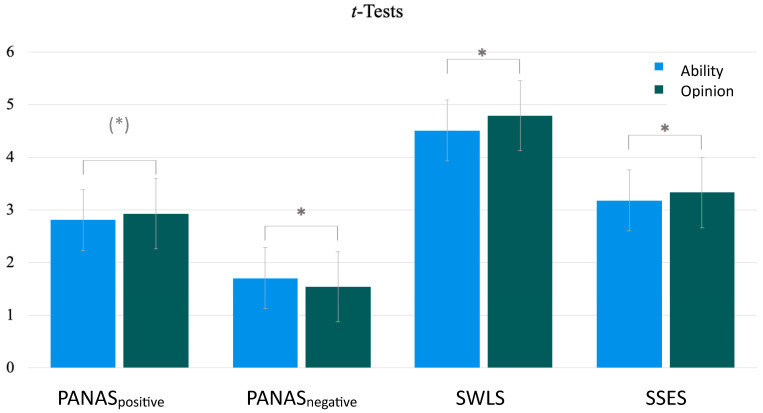
Differences in subjective well-being depending on experimental conditions. *Note.* PANAS_positive/negative_ = Positive and Negative Affect Schedule; SWLS = Satisfaction With Life Scale; SSES = State Self-Esteem Scale. * *p* < 0.05; (*) *p* < 0.10.

**Table 1 behavsci-13-00850-t001:** Descriptions of the used measures.

Measure	Range	Total (*N* = 409) *M* (*SD*)	EG_Ability_ (*N* = 198) *M* (*SD*)	EG_Opinion_ (*N* = 211) *M* (*SD*)	*t*	*p*	*d*
PANAS_positive_	1–5	2.87 (0.72)	2.81 (0.71)	2.93 (0.72)	−1.84	0.067	0.71
PANAS_negative_	1–5	1.62 (0.75)	1.70 (0.82)	1.54 (0.68)	2.12	0.036	0.75
SWLS	1–7	4.66 (1.27)	4.51 (1.30)	4.79 (1.21)	−2.22	0.027	1.26
SSES	1–5	3.26 (0.72)	3.18 (0.74)	3.33 (0.69)	−2.03	0.043	0.72

*Note. dfs* = 407. EG_Ability_
*= Experimental Group Ability;* EG_Opinion_ = *Experimental Group Opinion*; *M* = mean; *SD* = standard deviation; *t* = *t*-Test of both groups; *d* = Cohens *d*; PANAS_positive/negative_ = Positive and Negative Affect Schedule; SWLS = Satisfaction With Life Scale; SSES = State Self-Esteem Scale. All scales were decoded so that higher values meant higher agreement.

**Table 2 behavsci-13-00850-t002:** Intercorrelations.

	1	2	3
1. PANAS_positive_	-		
2. PANAS_negative_	−0.197 ***	-	
3. SWLS	0.412 ***	−0.448 ***	-
4. SSES	0.428 ***	−0.518 ***	0.656 ***

Note. PANAS_positive/negative_ = Positive and Negative Affect Schedule; SWLS = Satisfaction With Life Scale; SSES = State Self-Esteem Scale; *** *p* < 0.001. When additional partial correlations were performed, taking into account the group membership, no significant difference was found with regard to the correlation variables.

## Data Availability

The dataset generated and analyzed during the current study can be reviewed at: https://osf.io/6rdvt/?view_only=d054f2297f1144c188753b2fe5644c06 (accessed on 30 August 2023).

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
