# Peer review of "The Impact of Social Comparisons More Related to Ability vs. More Related to Opinion on Well-Being: An Instagram Study"

_behavsci, 2023, doi:10.3390/bs13100850_

Round 1

Reviewer 1 Report

Overall, this is an interesting take in an area that is of growing importance.

A few minor issues: 

1. When presenting gender (2.3.1), a consistent number of decimal places should be used, and the last term ("diverse") should be clarified--I think you mean "gender diverse" or "non-binary."

2. In 2.3.2., you have a redundancy ("a manipulation check was included for the manipulation check was included").

Minor usage errors.

Author Response

Thank you so much for your comments! You will find our response attached!

Reviewer 2 Report

Comparing our abilities on Instagram makes us unhappy An exposition study with respect to ability vs. opinion-based social comparisons on Instagram

Title and Abstract

Overall, the paper seems to tackle an interesting and relevant topic. However, the abstract could benefit from minor revisions to improve clarity, academic tone, and logical flow.

  • The title is informative but could be streamlined for clarity and impact.
  • The abstract provides a good overview but suffers from minor language issues and could improve in terms of logical flow.
  • Keywords are relevant but could be expanded for better discoverability.

Academic English Writing

Title

The title is fairly clear but could be more concise. Consider revising it to something like: "The Impact of Ability vs. Opinion-Based Social Comparisons on Wellbeing: An Instagram Study."

Abstract

  • The term "experiencing great popularity" is somewhat informal. Consider using "gaining widespread popularity" for a more academic tone.
  • "On these platforms, users are continuously exposed to self relevant information" could be clearer as "On these platforms, users are continuously exposed to self-relevant information."
  • "an online exposition experiment (N = 409) was con ducted" has a spacing issue. It should be "conducted."
  • "adequate exposition stimulus material was selected in advance" is vague. What do you mean by "adequate"? This needs clarification.
  • "Theoretical and practical implications of the findings were discussed and suggestions for future work were outlined." This sentence is a bit generic. It would be more impactful to briefly state what some of those implications and suggestions are.

Introduction

  • The phrase "Facebook, Twitter, Instagram with up to 2.7 billion active users per month," lacks clarity. Consider revising to: "Facebook, Twitter, and Instagram collectively boast up to 2.7 billion active users per month."
  • "SNSs are about people. Therefore, use of SNSs is regularly connected with the elicitation of social comparisons." This sentence is a bit informal and has a spacing issue. Consider revising to: "Therefore, the use of SNSs is intrinsically linked to the elicitation of social comparisons."
  • "A highly relevant question in terms of personal, organizational, and societal considerations is what are the consequences of such comparisons on the users’ wellbeing?" This sentence is awkwardly phrased. Consider: "A question of significant relevance, both personally and societally, is: what impact do these comparisons have on users' wellbeing?"

Discussion and Conclusions

Critique and Suggestions:

By addressing the following points, the paper could offer a more comprehensive, clear, and impactful discussion of its implications, thereby enhancing its overall quality and usefulness.

  1. Explicit Implications Section:
    • Clarity and Cohesion: The text is dense and could benefit from better organization. Subheadings could be used to separate different points for easier navigation.
    • Critique: The paper integrates implications within the Discussion section but lacks a dedicated "Implications" section. This makes it harder for readers to pinpoint the practical and theoretical ramifications of the study.
    • Suggestion: Create a separate "Implications" section that succinctly outlines the practical and theoretical consequences. This will not only improve readability but also emphasize the importance of the study's findings.
    • Unexplored Factors: The discussion hints at other factors like the direction of comparison (upward or downward) but doesn't elaborate. Future research directions should be more explicitly stated.

  1. Clarity and Depth in Discussing Implications:
    • Critique: The paper mentions the implications but doesn't delve into them in detail. For instance, it talks about the relevance for clinical psychology but doesn't specify how clinicians could use this information.
    • Suggestion: Expand on each implication by providing concrete examples or recommendations. For instance, if the study is relevant for clinical psychology, how might a therapist use this information in treatment planning?
  2. Broadening the Scope of Implications:
    • Critique: The paper focuses mainly on the implications for clinical psychology and social media design but doesn't explore broader societal implications.
    • Suggestion: Discuss how the findings could influence public policy, educational programs, or even corporate responsibility. For example, could these findings inform guidelines for ethical design in social media platforms?
  3. Theoretical Implications:
    • Critique: While the paper does touch on how the findings relate to existing theories like social comparison theory, it doesn't fully explore how the study might challenge or extend these theories.
    • Suggestion: Discuss how the findings align or contrast with existing theories and what new questions or hypotheses they might generate. This could be particularly relevant for future research directions.
  4. User-Centric Implications:
    • Critique: The paper talks about the "Wellbeing Team" on Instagram but doesn't discuss how individual users could benefit from the findings.
    • Suggestion: Include recommendations or guidelines for end-users on how to navigate social media in a way that is less detrimental to their mental health based on the study's findings.
  5. Limitations and Implications:
    • Critique: The "Limitations" section is thorough but doesn't tie these limitations back to the implications. Understanding the limitations can help qualify the implications.
    • Suggestion: Briefly discuss how the limitations of the study might affect the generalizability or applicability of the implications. For example, since the sample was not diverse in age, caution should be exercised when extending the implications to older populations.

See attached file.

Author Response

Thank you for your sympathetic and constructive suggestions for improvement! It helped us a lot to improve our manuscript! You will find our revision attached!

Reviewer 3 Report

Dear authors,

Thank you for the interesting paper that focuses on the distinction between abilities and opinions in social media.

Festinger (1954) made the difference between ability and opinion clear by pointing out that abilities are “measurable in terms of objective performance criteria." When looking at your stimuli (Appendix A3), I am wondering if there are objective performance criteria. Could you elaborate what are the objective performance criteria for each stimuli? If you can describe the criteria, then we are not dealing with opinions.

Moreover, your images show hobbies, such as playing guitar and making drawings, suggesting an alternative interpretation than abilities and their objective performance criteria. It is most likely due to your research setting which causes participants to choose from ability and opinion. However, the participants made decisions on whether each stimulus was “more related to abilities or opinions,” thus they did not choose the ability or opinion. To be fair, the title of the article and Discussion should reflect that fact somehow. Consider the title:  “... study with respect to “more related to ability”- vs. “more related to opinion”-based social comparisons…”

Do you have permission from Meta to publish e.g. a faked Instagram logo? The local ethical committee of XXX supported this type of plagiarism? 

Your version:

Instagram logo 2018:

Other observations:

PDF page numbers.

pp.1-2: no longer Twitter but X

p.5: When and where was the data collected?

p.5: Could you be more detailed with the procedure? What happened with the qualitative materials collected from free-text fields? Could you analyze and report those findings as well?

p.6: Please show the demographic results: education and the number of followers.

p.6: Give some examples of the Affect Schedule.

p.6: Please describe what we have in Table 1. What is EGAbility/EGOpinion? You could also point out that higher numbers in the SWLS scale are negative (i.e., 7 = strongly disagree).

p.6: Control variables (gender, age, education, number of followers) should be tested.

p.7: Your variance estimates are based on … t-tests?

p.7, Table 2: How were the experimental groups doing?

p7. Here you have “PANASpositive and PANASnegative”: In Table 1, it was “PANASpositiv and PANASnegativ”. Please be consistent across the paper!

p.4: “subjective well-being Is this” (period is missing)

p.5: “manipulation check was included for the manipulation check was included”

p.8: “this issue. such”

Appendix B & C, please translate in English (Fokus, Ngesamt)

Author Response

(The authors gave the same response as above.)

Round 2

Reviewer 2 Report

The authors have successfully revised the paper!

Author Response

Thank you so much for your feedback!

Reviewer 3 Report

Dear authors,

The paper has improved from the previous version and the authors have answered my questions and suggestions. Proofreading is recommended before publishing the paper because there are some typos, difficult sentence structures and other stylistic issues that the proofreader can detect. I point out some of those below.

I wish all the best for your project!

p.1 Abstract. Introduce the notion before the abbreviation (i.e. SNSs).

p.1 You are defining your key concept, SNSs, no less than three times on this page. Perhaps you could define the abbreviation in your Abstract and then once in Introduction.

p.1 The title of section 1.1. is “(SNS)” - not sites?

p.3 “In, stable global self-worth...” In what? Perhaps “In general,...”

p.8 Figure 1, “(*)” is located too left 

p.9 “One factor … might be that ability-related social comparisons contrast with opinion-related social comparisons_is_automatically instigated.”

p.9 “Our findings expand Festingers [12]” => Festinger’s

p.10 “imaged-based SNSs” or rather “image-based SNSs”

Author Response

Thank you so much for your feedback! We proofread the paper and revised all tipos as well as other small mistakes we found! Thanks.